# IL-1β-Induced CXCL10 Expression in THP-1 Monocytic Cells Involves the JNK/c-Jun and NF-κB-Mediated Signaling

**DOI:** 10.3390/ph17070823

**Published:** 2024-06-22

**Authors:** Shihab Kochumon, Amnah Al-Sayyar, Texy Jacob, Hossein Arefanian, Fatemah Bahman, Nourah Almansour, Fawaz Alzaid, Fahd Al-Mulla, Sardar Sindhu, Rasheed Ahmad

**Affiliations:** 1Immunology & Microbiology Department, Dasman Diabetes Institute, Dasman 15462, Kuwait; shihab.kochumon@dasmaninstitute.org (S.K.); texy.jacob@dasmaninstitute.org (T.J.); hossein.arefanian@dasmaninstitute.org (H.A.); fatemah.bahman@dasmaninstitute.org (F.B.); nourah.almansour@dasmaninstitute.org (N.A.); sardar.sindhu@dasmaninstitute.org (S.S.); 2Centre d’Immunologie de Marseille-Luminy, Aix Marseille Université, Inserm, 13288 Marseille, France; alsayyar@ciml.univ-mrs.fr; 3Bioenergetics & Neurometabolism Department, Dasman Diabetes Institute, Dasman 15462, Kuwait; fawaz.alzaid@dasmaninstitute.org; 4Institut Necker Enfants Malades (INEM), INSERM U1151/CNRS UMRS8253, IMMEDIAB, Université deParis Cité, 75015 Paris, France; 5Translational Research Department, Dasman Diabetes Institute, Dasman 15462, Kuwait; fahd.almulla@dasmaninstitute.org; 6Animal & Imaging Core Facilities, Dasman Diabetes Institute, Dasman 15462, Kuwait

**Keywords:** IL-1β, CXCL10, JNK/c-Jun, NF-κB, THP-1 monocytic cells

## Abstract

CXCL10 (IP-10) plays a key role in leukocyte homing to the inflamed tissues and its increased levels are associated with the pathophysiology of various inflammatory diseases including obesity and type 2 diabetes. IL-1β is a key proinflammatory cytokine that is found upregulated in meta-inflammatory conditions and acts as a potent activator, inducing the expression of cytokines/chemokines by immune cells. However, it is unclear whether IL-1β induces the expression of CXCL10 in monocytic cells. We, therefore, determined the CXCL10 induction using IL-1β in THP1 monocytic cells and investigated the mechanisms involved. Monocytes (human monocytic THP-1 cells) were stimulated with IL-1β. CXCL10 gene expression was determined with real-time RT-PCR. CXCL10 protein was determined using ELISA. Signaling pathways were identified by using Western blotting, inhibitors, siRNA transfections, and kinase assay. Our data show that IL-1β induced the CXCL10 expression at both mRNA and protein levels in monocytic cells (*p* = 0.0001). Notably, only the JNK inhibitor (SP600125) significantly suppressed the IL-1β-induced CXCL10 expression, while the inhibitors of MEK1/2 (U0126), ERK1/2 (PD98059), and p38 MAPK (SB203580) had no significant effect. Furthermore, IL-1β-induced CXCL10 expression was decreased in monocytic cells deficient in JNK/c-Jun. Accordingly, inhibiting the JNK kinase activity markedly reduced the IL-1β-induced JNK/c-Jun phosphorylation in monocytic cells. NF-κB inhibition by Bay-117085 and resveratrol also suppressed the CXCL10 expression. Our findings provide preliminary evidence that IL-1β stimulation induces the expression of CXCL10 in monocytic cells which requires signaling via the JNK/c-Jun/NF-κB axis.

## 1. Introduction

Diabesity represents the co-existence of obesity and type 2 diabetes (T2D) which has escalated to a global epidemic level. It is associated with a constellation of metabolic disorders marked by insulin resistance and hyperinsulinemia [1,2]. In diabesity, an increase in the adipose tissue size harbingers perturbations in energy metabolism neuroendocrine and immune functions [3]. The chronic low-grade inflammation which is considered a hallmark of diabesity acts as a driver of immune dysregulation, resulting from an increased expression of proinflammatory cytokines and chemokines. Chemokines are chemotactic cytokines that regulate the directional movements of leukocytes, endothelial, and epithelial cells and play roles in inflammation, activation of host immune responses, and other biological processes including host defense, apoptosis, cell growth and proliferation, angiogenesis, and hematopoiesis [4].

C-X-C motif chemokine ligand 10 (CXCL10), also known as interferon gamma-inducible protein 10 (IP-10), is a proinflammatory chemokine that regulates immune responses through the recruitment and activation of leukocytes and is secreted by a variety of cells including monocytes/macrophages [5], neutrophils [6], fibroblasts [7], and endothelial cells. Its inflammatory role has been reported in the pathophysiology of several diseases such as liver disease [8,9,10], multiple sclerosis [11], obesity, and type 2 diabetes [12,13]. Elevated CXCL10 levels in diabetic patients were found to be associated with an increased risk of inflammation and insulin resistance [12]. Furthermore, D’Esposito et al. reported increased CXCL10 and IL-1β secretion in adipocytes from obese individuals that was linked to impaired glucose tolerance [14].

IL-1β is known to induce CXCL10 expression in a variety of cells, such as preadipocytes [15] as well as human pancreatic β-cells and endometrial stromal cells [16,17]. Monocytes play a key role in metabolic inflammation in individuals with diabesity, and these individuals also display increased circulatory levels of IL-1β and the chemokine CXCL10. The extravasation of activated monocytes from circulation into the tissues such as adipose tissue, liver, and pancreas occurs during the disease progression and leads to inflammatory changes within these tissues of metabolic significance [18]. Since the role of IL-1β in CXCL10 expression in THP-1 monocytic cells remains unclear, in this study, we tested the hypothesis of whether IL-1β could induce CXCL10 expression in monocytic cells, and, if so, which molecular mechanism was involved. Herein, we show the evidence that IL-1β induces the expression of CXCL10 in THP-1 monocytic cells through the mechanism involving JNK/c-Jun-mediated signaling.

## 2. Results

### 2.1. IL-1β Treatment Upregulates CXCL10 Gene and Protein Expression in THP-1 Cells

Since IL-1β is a strong immune cell stimulator known to induce the expression of inflammatory cytokines, we sought to determine the expression of CXCL10 in THP-1 cells challenged with IL-1β. In this regard, we found that CXCL10 mRNA expression was significantly higher (16.63 ± 1.75 fold; *p* = 0.0001) in IL-1β-treated THP-1 cells than controls (Figure 1A). As expected, CXCL10 secreted protein levels were also found to be higher in supernatants from IL-1β-stimulated THP-1 cells (198 ± 16.7 pg/mL) compared to controls (13.33 ± 3.21 pg/mL; *p* = 0.0001) (Figure 1B). Considering the functional significance of macrophages in adipose tissue inflammation, we further asked whether IL-1β could also induce CXCL10 production in macrophages. Using THP–1–derived macrophages, we found that IL-1β enhances CXCL10 production (Appendix A).

### 2.2. Inhibiting JNK Suppresses the IL-1β-Induced CXCL10 Expression

Since the JNK (p38) and ERK MAPK kinases are activated in cells that are challenged with inflammatory cytokines, we sought to determine if these kinases were involved in CXCL10 upregulation in IL-1β-stimulated monocytic cells. To test this, THP-1 cells were first treated with inhibitors of MEK1/2 (U0126), ERK1/2 (PD98059), JNK (SP600125), and p38 (SB203580) and then stimulated with IL-1β. We found that the inhibition of JNK signaling significantly suppressed the gene/protein expression of CXCL10 in IL-1β stimulated THP-1 cells, whereas the inhibition of p38, ERK1/2, and MEK1/2 signaling had a non-significant effect (Figure 2A,B).

### 2.3. JNK Deficiency Also Leads to Suppression in IL-1β-Induced CXCL10

To further verify that IL-1β-induced CXCL10 expression was JNK-dependent, THP-1 cells were transfected with JNK siRNA, leading to a significant reduction in *JNK* mRNA levels in transfected cells, compared to control transfected with scrambled siRNA (Figure 3A). As expected, siRNA-mediated JNK deficiency led to a significant diminution in IL-1β-induced CXCL10 expression at both transcriptional and translational levels (Figure 3B,C), further corroborating that this induction was JNK-dependent.

### 2.4. IL-1β-Induced CXCL10 Expression in THP-1 Cells Involves c-Jun Mediated Signaling

Given that JNK activation, in turn, leads to phosphorylation and the activation of its major downstream substrate c-Jun, defective c-Jun signaling is also expected to suppress the expression of CXCL10 in IL-1β-challenged monocytic cells. To verify this, THP-1 cells were transfected with c-Jun siRNA, resulting in a significant decrease in *c-Jun* mRNA expression, compared to control transfected with scrambled siRNA (Figure 4A). As anticipated, the c-Jun deficiency also led to reduced CXCL10 expression in response to IL-1β stimulation (Figure 4B,C), confirming the involvement of the c-Jun kinase signaling cascade.

### 2.5. IL-1β Stimulation Induces JNK/c-Jun Phosphorylation and Enhances the JNK Kinase Activity in THP-1 Cells

We next assessed the effect of IL-1β stimulation on JNK phosphorylation and kinase activity in THP-1 cells. To this effect, Western blotting shows that IL-1β stimulation-induced phosphorylation of JNK and c-Jun in THP-1 cells (Figure 5A–C). Further, to determine changes in JNK kinase activity, cell lysates were incubated overnight with c-Jun fusion protein beads, and the kinase assay was performed. Our data show that the JNK kinase activity was significantly enhanced after THP-1 cell stimulation with IL-1β, compared to activity in control lysates (Figure 5D,E).

### 2.6. IL-1β-Induced CXCL10 Expression Also Involves the Canonical NF-κB-Mediated Signaling

IL-1β is known to activate the canonical NF-κB pathway. To determine whether this activation was also involved in CXCL10 expression in monocytic cells, NF-κB pathway inhibitors (Bay 11-7085 and resveratrol) were used to treat THP-1 cells before IL-1β stimulation. As anticipated, pretreatment with NF-κB inhibitors led to a significant suppression in both CXCL10 mRNA and secreted protein, compared to controls (Figure 6A,B). In addition, IL-1β treatment induced the NF-κB phosphorylation in a time-dependent manner (Figure 6C,D).

## 3. Discussion

Adipose expression of CXCL10 is upregulated in metabolic conditions, such as diabesity, which plays a critical role in adipose tissue inflammation, insulin resistance, and metabolic dysfunction through changes that involve cellular extravasation into adipose tissue and the activation of major immune effector cells such as monocytes/macrophages, T-cells, dendritic cells, and NK cells [19]. Previously, we and others reported that the increased adipose expression of CXCL10 in individuals with obesity was associated with inflammatory (IL-1β expression, C-reactive protein levels) and other factors (BMI, CXCR3, and reduced neovascularization causing adipose tissue hypoxia) [19,20,21]. Among the various immune effector cell populations involved, monocytes/macrophages are of prime significance, and the fundamental question as to which molecular mechanism(s) may drive the expression of CXCL10 in monocytic cells remains to be addressed. Our data clearly show that IL-1β stimulation promotes the expression of CXCL10 in THP-1 monocytic cells at both the transcriptional and translational levels. These findings are consistent with those of Burke et al. showing that a synergistic interaction between two inflammatory cytokines IL-1β and IFN-γ leads to the upregulation of *CXCL10* gene expression in rat islets and β-cell lines [22]. Likewise, it was shown that various stimuli induced and promoted CXCL10 expression in both immune and non-immune cells [23,24].

Next, we addressed the signaling mechanism that regulated CXCL10 expression in IL-1β-stimulated THP-1 cells. By pre-treating cells with specific pharmacological inhibitors, we identified that MAPKs were pivotally involved in IL-1β-induced CXCL10 expression in monocytic cells. As per these findings, CXCL10 expression involved the JNK-mediated signaling since this expression was significantly suppressed at the transcriptional and translational levels in cells that were treated with JNK inhibitor SP600125 before stimulating with IL-1β. While the treatments with other inhibitors such as U0126 (MEK1/2 inhibitor), PD98059 (ERK1/2 inhibitor), and SB203580 (p38 inhibitor) rather led to non-significant increases in CXCL10 expression, suggesting that these pathways were not involved in CXCL10 expression, and it appeared that the blockade of signaling through these pathways could have a non-specific, albeit statistically non-significant, stimulatory effect on CXCL10 production which may be due to the activation of other regulatory pathways that could stabilize the CXCL10 gene expression. These data show that the JNK signaling was critical to the induction of CXCL10 in monocytic cells challenged with IL-1β. MAPKs are known to regulate the expression of various chemokines in monocytic cells; however, which MAPK is involved depends on the stimulus type and the chemokine produced. Indeed, our finding regarding the JNK involvement is in line with a previous study indicating that CXCL10 production by airway smooth muscle cells was dependent on MAPK-JNK-mediated signaling, whereby ERK was not involved and p38 only mediated the CXCL10 expression co-induced via TNF-α and IL-1β in these cells [25]. Using colorectal cancer cell lines (Caco-2 and HT29 cells), Sunil et al. demonstrated that IL-1β-induced CXCL10 mRNA/protein expression was dependent on NF-κB, ERK1/2, and p38 MAPK-mediated signaling. Consistent with our results, it was also noted that the ERK1/2 inhibitor PD98059 led to an increase in CXCL10 mRNA expression in H29 cells; however, not at the protein level [26].

To further validate that JNK was involved, the siRNA-mediated JNK suppression approach was used, and, as expected, the JNK deficiency resulted in a significant reduction in CXCL10 gene and protein expression in THP-1 cells. Based on a positive regulatory role of JNK in CXCL10 induction, we further surmised a similar role of c-Jun in these cells. To this end, and as expected, the c-Jun deficiency in these cells also led to a significant reduction in the CXCL10 gene/protein expression. Together, these data support that both JNK and c-Jun regulated the expression of CXCL10 in THP-1 monocytic cells. Notably, c-Jun activity is regulated by the NH_2_-terminal phosphorylation of its transactivation domain by the JNK family kinases which form a stable complex with c-Jun. Our data shed further light on the c-Jun/JNK-mediated co-regulation of *CXCL10* in monocytic cells, indicating that both these transcription factors were implicated in IL-1β-induced CXCL10 expression in these cells. Consistent with our findings, Holzberg et al. showed that the c-Jun/JNK complex was involved in the regulation of a distinct set of IL-1β-induced inflammatory genes [27]. The c-Jun is one of the best-characterized components of the transcription factor activator protein (AP)-1 and the binding of JNK to c-Jun forms a high-affinity signaling complex [28], which does not require the JNK catalytic activity or presence of phospho-acceptor sites in c-Jun [29]. Overall, the c-Jun/JNK interaction has two main purposes: first, it provides the specificity of JNK for c-Jun, and, second, it elevates the JNK concentration at the gene promoters that bind c-Jun, thus upregulating c-Jun-mediated transcription. Of note, mammalian JNKs were originally named stress-activated protein kinases (SAPK) as they were found to be activated by different stress stimuli including growth factors and proinflammatory cytokines, which aligns with our study showing JNK activation in monocytic cells in response to IL-1β stimulation. Furthermore, we found that the SAPK/JNK kinase activity was elevated in CXCL10-expressing THP-1 cells that were challenged with IL-1β, which is corroborated by other studies supporting the activation of JNK/c-Jun by TNF-α and/or IL-1β [30,31]. Notably, our findings showing the diminished monocytic production of CXCL10 following treatment with the JNK inhibitor (SP600125) or transfection with JNK-/c-Jun-specific siRNAs may have clinical significance and the potential to be used as siRNA therapeutics, as in vivo RNA interference (RNAi) for targeted gene silencing may be an effective strategy to reduce the systemic levels of CXCL10 and alleviate associated metabolic inflammation. Further validation studies in suitable animal models will be highly desirable.

Next, in addition to the involvement of the JNK/c-Jun pathway, we also assessed the role of canonical NF-κB signaling in this CXCL10 expression induced by IL-1β stimulation in THP-1 cells. As expected, and hinged on two key observations, our data clearly show that the canonical pathway is also involved in CXCL10 induction. First, the cell pre-treatment with two NF-κB inhibitors, such as Bay117085 and resveratrol, significantly quenched the expression of CXCL10. Second, IL-1β stimulation of THP-1 cells induced the NF-κB phosphorylation in a time-dependent manner. Indeed, NF-κB is one of the most important intracellular nuclear transcription factors that plays a key role in the transcriptional regulation of several genes, including those of cytokines/chemokines [32]. These data also suggest that NF-κB inhibition by Bay117085 and resveratrol might have therapeutic potential regarding the suppression of CXCL10 in metabolic disorders, such as obesity/T2D, which involve systemic increases in inflammatory cytokine IL-1β. NF-κB is a master regulator of inflammation and the NF-κB pathway is the major proinflammatory signaling pathway that regulates transcriptional activation of responsive cytokine and chemokine genes [33,34]. Consistent with our findings, at least in part, Qi et al. showed that TNF-α-induced NF-κB activation might be the primary pathway in THP-1 cells that contributed to the production of CXCL10. They also showed that IFN-γ further potentiated TNF-α-induced CXCL10 expression by activating STAT1 and NF-κB through JAK1/2 signaling [35].

However, the present study has certain limitations. First, the responsiveness to cytokines might depend on the state of cellular differentiation and, therefore, future studies might also use primary monocytes/macrophages to verify and extrapolate the current findings. At the same time, it is noteworthy that multiple factors like culture conditions, purity, types of contaminating cells present or heterogeneity of cultures, and the cellular stress inflicted by isolation/purification procedures can, most likely, influence the data obtained from primary cells. Since we used THP-1 cells, which are from a human cancer (monocytic leukemia) cell line, there still remains a limitation and a need for using primary human monocytes/macrophages to verify these preliminary data. Second, it would also be interesting to study the comparative induction of CXCL10 through numerous inflammatory stimuli that are clinically relevant to acute or chronic inflammation such as IFN-γ, TNF-α, IL-6, CCL2, and CXCL8 in key immune effector cells. We speculate that studying the synergistic interactions among such inflammatory triggers for CXCL10 induction/upregulation in major immune effectors would be another interesting scenario, given the emerging pathophysiological significance of these inflammatory proteins in metabolic disorders. Therefore, further studies will be required to shed light on these open questions.

## 4. Materials and Methods

### 4.1. Cell Culture and Stimulations

THP-1 human monocytic cells were grown in RPMI-1640 culture medium supplemented with 10% fetal bovine serum, 2 mM glutamine, 1 mM sodium pyruvate, 10 mM HEPES, 100 ug/mL normocin, 50 U/mL penicillin, and 50 μg/mL streptomycin and incubated at 37 °C (with humidity) in 5% CO_2_. For stimulation, cells were cultured in triplicate wells of 12-well plates (Costar, Corning Incorporated, Corning, NY, USA) at 1 × 10^6^ cells/well concentration unless indicated otherwise. Cells were stimulated with IL-1β (10 ng/mL) for 24 h at 37 °C. Later, the cells were harvested for total RNA isolation for gene expression analysis using RT-qPCR, and the culture media were collected, clarified by centrifugation, aliquoted, and stored at −80 °C until use for measuring CXCL10-secreted protein concentrations.

### 4.2. Small Interfering RNA (siRNA) Transfections

For transient transfection, THP-1 cells (1 × 10^6^ cells) were resuspended in 100 μL of nucleofector solution provided with the Amaxa Nucleofector Kit V and transfected separately with SAPK/JNK siRNA, c-Jun siRNA, and Negative Control (scrambled) siRNA. All transfections were performed with Amaxa Cell Line Nucleofector Kit V for THP-1 (Lonza, Cologne, Germany) by using the Amaxa Electroporation System (Amaxa Inc., Cologne, Germany). After 36 h of transfection, cells were treated with IL-1β (10 ng/mL) for 24 h. Cells were harvested for total RNA isolation and culture media were collected for measuring CXCL10 secretion. SAPK/JNK and c-Jun gene knockdown levels were assessed using real-time RT-PCR using gene-specific TaqMan assays.

### 4.3. Real Time Reverse Transcription–Polymerase Chain Reaction (RT-PCR)

Total RNA was extracted from treated and untreated THP-1 cells using RNeasy Mini Kit (Qiagen, Valencia. CA, USA). Total RNA (1 μg) was used for cDNA conversion using a high-capacity cDNA reverse transcription kit (Applied Biosystems, Foster City, CA, USA). For real time-PCR, 50 ng of cDNA was used with gene-specific TaqMan Gene Expression Assay products (IP-10, Hs00171042_m1; c-Jun, Hs01103582_s1; JNK, Hs01548508_m1; and GAPDH, Hs03929097_g1) and TaqMan^®^ Gene Expression Master Mix (Thermo Scientific/Applied Biosystems, Foster City, CA, USA) in a QuantStudio 5 Fast Real-Time PCR System (Applied Biosystems, Foster City, CA, USA).

All the target Ct values were normalized with corresponding GAPDH Ct values and the expression levels of CXCL10 in treated samples relative to control samples were calculated using the 2^–∆∆Ct^ method. Relative mRNA expression was expressed as fold expression over average control gene expression. The expression level in the control treatment was assumed to be one.

### 4.4. SAPK/JNK Kinase Assay

SAPK/JNK kinase activity was determined using the SAPK/JNK Kinase assay kit (Cell Signaling Technology (CST), Danvers, MA, USA) according to the manufacturer’s instructions. Briefly, cell lysates from IL-1β-treated and control samples were prepared using lysis buffer, and immunoprecipitation of phospho-SAPK/JNK was carried out using immobilized phospho-SAPK/JNK rabbit mAb linked to agarose beads to pull down SAPK/JNK kinases from cell extracts. The immunoprecipitated SAPK/JNK extract was incubated with the c-Jun fusion protein, kinase buffer, and ATP. The c-Jun phosphorylation mediated using SAPK/JNK kinase was measured via immunoblotting using phospho c-Jun antibody.

### 4.5. Quantification of Secreted CXCL10 Protein

Secreted CXCL10 protein in culture supernatants of IL-1β-stimulated and vehicle-treated (controls) THP-1 monocytic cells was quantified using sandwich ELISA, following the manufacturer’s instructions (R&D systems, Minneapolis, MN, USA).

### 4.6. Western Blotting

THP-1 cells treated with IL-1β at different time points were harvested and incubated for 30 min with RIPA lysis buffer for total protein extraction. The lysates were centrifuged at 14,000× *g* for 10 min and the supernatants were collected. Protein concentration was measured by using QuickStart Bradford Dye Reagent, 1x Protein Assay kit (Bio-Rad Laboratories, Inc., Hercules, CA, USA). Protein (20 μg) samples were mixed with loading buffer, heated for 5 min at 95 °C, loaded onto the gel, and resolved by 12% SDS-PAGE. Fractionated proteins were transferred to the Immuno-Blot PVDF membrane (Bio-Rad Laboratories, Hercules, CA, USA) via electro blotting. The membranes were blocked with 5% non-fat milk in PBS for 1 h, followed via incubation with primary antibodies against p-SAPK/JNK (Cat# 9251), SAPK/JNK (Cat# 9252), p-c-Jun (Cat# 3270), C-Jun (Cat# 9165), p-NF-κB (Cat# 3031), NFKB (Cat# 8242), and β-actin (Cat# 4967) (1:1000 dilution) at 4 °C overnight. All primary antibodies were purchased from Cell Signaling Technology (CST). The blots were then washed three times with TBS-T and incubated for 2 hrs with HRP-conjugated secondary antibody (Promega, Madison, WI, USA). Immunoreactive bands were developed using an Amersham ECL Plus Western Blotting Detection System (GE Health Care, Buckinghamshire, UK) and visualized using Molecular Imager^®^ VersaDoc™ MP Imaging Systems (Bio-Rad Laboratories, Hercules, CA, USA).

### 4.7. Statistical Analysis

Statistical analysis was performed using GraphPad Prism software 10.2.3(403) (La Jolla, CA, USA). Data are shown as mean ± standard error of the mean (SEM) unless otherwise indicated. Student t-test and one-way ANOVA were used to compare group means with that of the control group. For all analyses, *p* value < 0.05 was considered significant (*), representing *p* < 0.01 as highly significant (**), and *p* < 0.001/*p* < 0.0001 as extremely significant (***/****).

## 5. Conclusions

Taken together, our data support that IL-1β stimulation induces the expression of CXCL10 in THP-1 monocytic cells. Mechanistically, this CXCL10 induction involved the JNK/c-Jun and NF-κB-mediated signaling. Furthermore, the inhibition of JNK kinase activity reduced the IL-1β-induced phosphorylation of JNK and c-Jun. These findings provide the preliminary evidence that IL-1β is instrumental in inducing CXCL10 in monocytic cells which may have significance for chemokine-associated inflammation and metabolic dysregulation.

## Figures and Tables

**Figure 1 pharmaceuticals-17-00823-f001:**
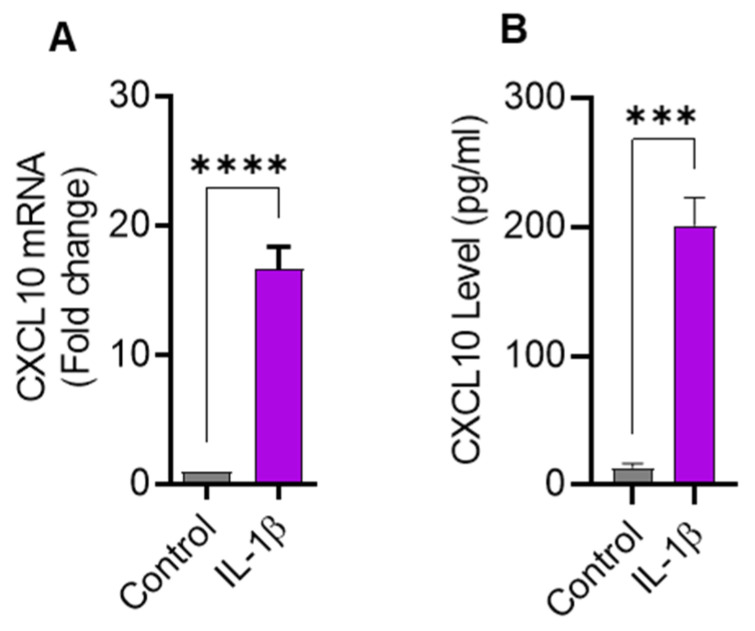
IL-1β stimulation induces the expression of CXCL10 in THP-1 human monocytic cells. THP-1 cells were cultured (1 × 10^6^ cells per well) in 12-well plates and cells were stimulated for 24 h with IL-1β (10 ng/mL) and cells treated with vehicle served as control. Total cellular RNA was isolated and used for CXCL10 mRNA expression using real-time RT-PCR and CXCL10 secreted protein in culture supernatants was determined using ELISA, per description in Materials and Methods. Expression of significantly increased CXCL10 (**A**) mRNA and (**B**) protein are shown in IL-1β-stimulated THP-1 monocytic cells compared with controls. All data are expressed as mean ± SEM values (*n* = 3). *** *p* ≤ 0.001, **** *p* ≤ 0.0001 is considered highly significant.

**Figure 2 pharmaceuticals-17-00823-f002:**
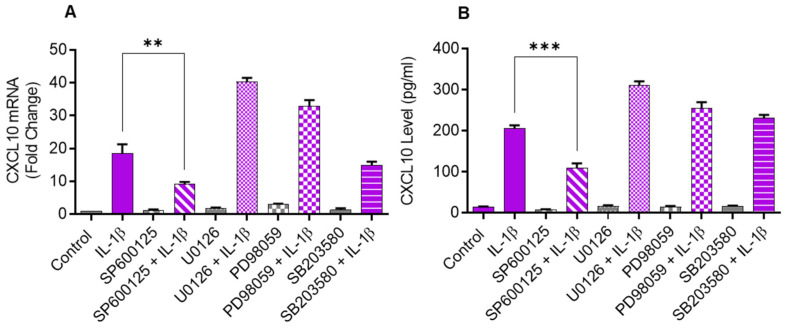
IL-1β-induced CXCL10 expression in THP-1 cells is dependent on JNK signaling. THP-1 cells were treated for 1 h with inhibitors of MEK1/2 (U0126; 10 µM), ERK1/2 (PD98059; 10 µM), JNK (SP600125; 10 µM), and p38 (SB203580; 10 µM), followed by stimulation with IL-1β (10 ng/mL) for 24 h. CXCL10 gene and protein expression were determined using real-time RT-PCR and ELISA, respectively, as stated in Materials and Methods. A significant suppression in CXCL10 (**A**) gene and (**B**) protein expression was observed in cells that were pre-treated with SP600125, compared to respective controls, suggesting that IL-1β-induced CXCL10 expression in THP-1 cells was dependent on JNK signaling. All data are expressed as mean ± SEM values (*n* = 3; ** *p* < 0.01, *** *p* < 0.001).

**Figure 3 pharmaceuticals-17-00823-f003:**
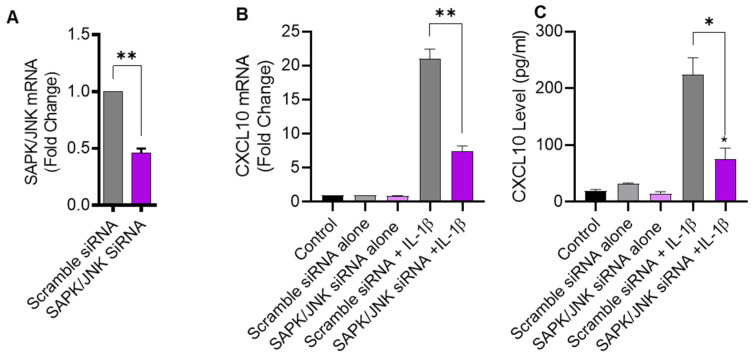
JNK deficiency attenuates the CXCL10 expression in response to IL-1β. THP-1 cells were transfected with JNK siRNA (20 nM) while control cells were transfected with scrambled siRNA (20 nM). After 36 h, transfected cells were stimulated for 24 h with IL-1β (10 ng/mL) and CXCL10 gene, and protein expression was determined using real-time RT-PCR and ELISA, respectively, as described in Materials and Methods. (**A**) A significant suppression in *SAPK*/*JNK* mRNA is shown in THP-1 cells after 36 h of transfection, compared to control. Consistent with reduced JNK expression, the IL-1β-induced CXCL10 gene (**B**) and protein (**C**) expression was diminished in THP-1 cells compared to respective controls. All data are expressed as mean ± SEM values (*n* = 3; * *p* < 0.05, ** *p* < 0.01).

**Figure 4 pharmaceuticals-17-00823-f004:**
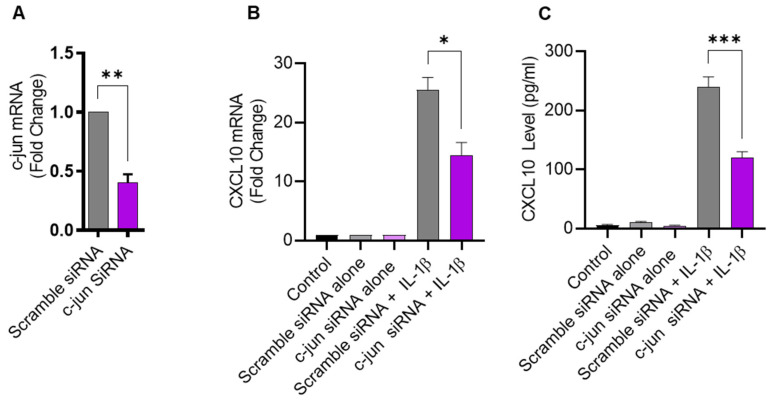
c-Jun deficiency attenuates the IL-1β-induced CXCL10 expression in monocytic cells. THP-1 cells were transfected with c-Jun siRNA (20 nM) and controls were transfected with scrambled siRNA (20 nM). After 36 h, transfected cells were stimulated for 24 h with IL-1β (10 ng/mL) and CXCL10 gene, and protein expression was determined using real-time RT-PCR and ELISA, respectively, as described in Materials and Methods. (**A**) A significant suppression in *c-Jun* mRNA is shown in THP-1 cells at 36 h post-transfection, compared to control. Consistent with the reduction in c-Jun expression, the IL-1β-induced CXCL10 gene (**B**) and protein (**C**) expression was significantly suppressed in THP-1 cells, compared to respective controls. All data are expressed as mean ± SEM values (*n* = 3). * *p* < 0.05, ** *p* < 0.01, *** *p* < 0.001.

**Figure 5 pharmaceuticals-17-00823-f005:**
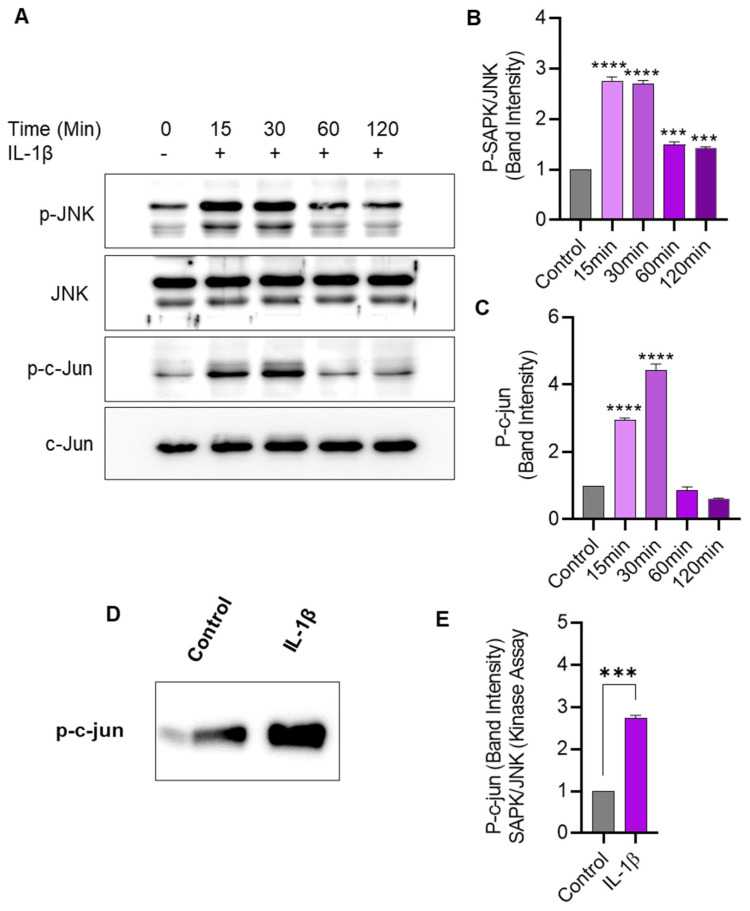
IL-1β stimulation induces JNK/c-Jun phosphorylation and SAPK/JNK kinase activity in monocytic cells. THP-1 cells were treated with IL-1β (10 ng/mL) for 15, 30, 60, and 120 min, and cell lysates were assessed for JNK/c-Jun phosphorylation using Western blotting and kinase activity was determined using SAPK/JNK kinase assay, as described in Materials and Methods. (**A**) Phosphorylated and total protein bands are shown for JNK and c-Jun at 0, 15, 30, 60, and 120 min after IL-1β stimulation. Densitometry of normalized data (phosphorylated to total protein band ratios) indicates that hyperphosphorylation of both JNK (**B**) and c-Jun (**C**) was detectable in monocytic cells at 15 min after IL-1β stimulation. (**D**) In the SAPK/JNK kinase assay, the c-Jun phosphorylation is shown in THP-1 cell lysates after stimulation with IL-1β, compared with controls that were treated with vehicle only. (**E**) Band densitometry analysis reveals a significant increase in SAPK/JNK kinase activity/c-Jun phosphorylation in IL-1β-treated cells, compared with control. All data are expressed as mean ± SEM values (*n* = 3; *** *p* < 0.001, **** *p* < 0.0001).

**Figure 6 pharmaceuticals-17-00823-f006:**
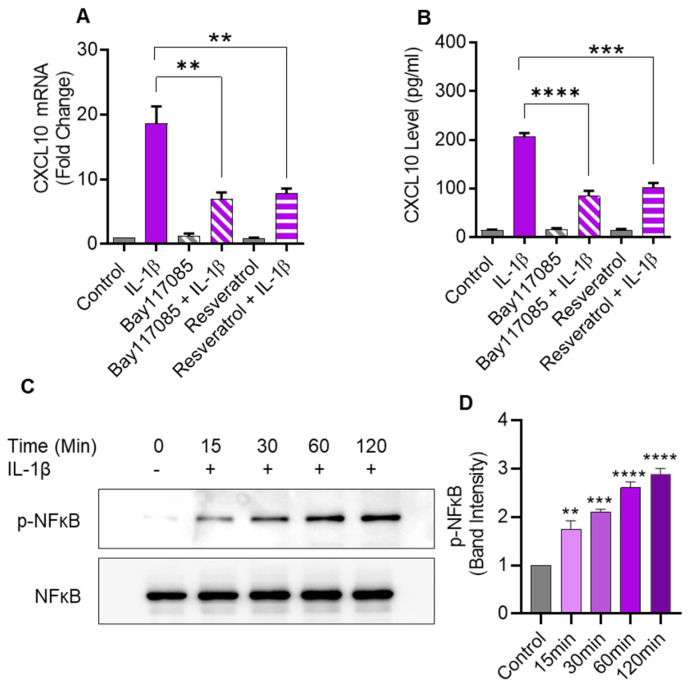
IL-1β-induced CXCL10 expression in human monocytic cells involves signaling via the canonical NF-κB pathway. THP-1 cells were pretreated for 1 h with NF-κB inhibitors such as Bay 11-7085 (10µM) or Resveratrol (10µM) and then stimulated for 24 hrs with IL-1β (10 ng/mL). CXCL10 mRNA and secreted protein were determined using real-time RT-PCR and ELISA, respectively, as described in Materials and Methods. A significant suppression in *CXCL10* mRNA (**A**) and CXCL10 secreted protein (**B**) expression was observed in IL-1β-stimulated cells that were pre-treated with NF-κB inhibitors, compared to likewise stimulated controls that were not treated with these inhibitors. (**C**) To assess IL-1β-induced NF-κB phosphorylation, THP-1 cells were incubated with IL-1β (10 ng/mL), and samples were collected at 0, 15, 30, 60, and 120 min, while the mock (control) was treated with vehicle-only. Cell lysates were resolved using SDS-PAGE and NF-κB phosphorylation was assessed using antibodies against phosphorylated and total NF-κB. Bands depict a time-dependent increase in NF-κB phosphorylation. (**D**) Band densitometry analysis of the ratios of phosphorylated to respective control (total NF-κB) band indicates a significant time-dependent upregulation of NF-κB phosphorylation in IL-1β-stimulated THP-1 cells over a time period of 15–120 min. All data are expressed as mean ± SEM values (*n* = 3; ** *p* < 0.01, *** *p* < 0.001, **** *p* < 0.0001).

## Data Availability

All data related to this study are presented in the publication.

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
