# Peer review of "IL-1β-Induced CXCL10 Expression in THP-1 Monocytic Cells Involves the JNK/c-Jun and NF-κB-Mediated Signaling"

_pharmaceuticals, 2024, doi:10.3390/ph17070823_

Round 1
Reviewer 1 Report
Comments and Suggestions for Authors
In this paper, the authors aim to investigate whether and how IL-1𝛽 induce CXCL10 expression in THP-1 monocytic cells, showing the evidence that IL-1𝛽 induces the expression of CXCL10 through activation of JNK/c-Jun signaling pathway. The pathophysiological context is that increased levels of IL-1𝛽-induced CXCL10 could be associated with the development or severity of obesity and type-2 diabetes. Data is interesting but not novel. However, if CXCL10 is expected to play a key role in leukocyte homing to the inflamed tissue, my first concern is the lack of explanation for the use of a monocytic cell line. One of the most important immune, inflammatory cell involved in the pathophysiology of inflammatory diseases is the macrophage, and in particular Kupffer cells in the adipose tissue and the liver. The authors should justify why they did not induce differentiation of THP-1 cells into macrophages (via PMA or otherwise). This would have made it possible by better mimicking in vivo conditions and link the results with the pathophysiological context. They should also clearly describe the limitations of the findings with a cancerous model that resembles peripheral monocytes.
Minor comments
Materials and Methods
1. The sequence of the primers used for qRT-PCR should be included in the manuscript, at least in the supplementary material.
2. The catalogue numbers of the antibodies used for Western Blotting must be included.
Results
Figure 2
1. Authors discuss the decrease in CXCL10 mRNA levels with JNK inhibitor, but there seems to be an increase of mRNA levels with U0126 (MEK1/2 inhibitor) in particular, and probably also using PD98059 (ERK1/2 inhibitor). Moreover, it is mentioned that the other inhibitors “had a non-significant effect” (lines 96-97). This statement is regarding the inhibition of CXCL10 only, or is the increase also non-significant? I think it could be interesting to describe/analyse and discuss these results.
2. In the figure legend, the concentration of PD98059 is missing (line 100).
Discussion
1. Comment for figure 2 could be discussed.
Novelty:
The link between IL-1b and CXCL10 is well established in other contexts. For example: Sunil et al. (2010) shows that IL-1b-induced CXCL10 mRNA expression and protein secretion are dependent on NFkB and ERK1/2 + p38 MAPK signaling in colorectal cancer cell lines (Caco-2 & HT29 cells). It also shows that inhibition of ERK1/2 (PD98059) causes an increase of CXCL10 mRNA in HT29 cells (also observed in THP-1 here) (https://link.springer.com/article/10.1007/s00384-009-0847-3)
In other independent study using THP-1 cells., Qi et al. (2009) shown that TNF-α induces CXCL10 production by activating NF-κB through ERK activation and that IFN-γ induces CXCL10 production by increasing the activation of STAT1 through JAKs pathways (https://onlinelibrary.wiley.com/doi/full/10.1002/jcp.21815).
Author Response
Response to Reviewer 1 Comments
Ref: Manuscript ID: pharmaceuticals-3004066
Title: IL-1β-induced CXCL10 expression in THP-1 monocytic cells involves the JNK/c-Jun and NF-κB mediated signaling
We thank the reviewer for thoughtful comments. Please see below our point by point response to the comments or concerns raised.
Comments and Suggestions for Authors
In this paper, the authors aim to investigate whether and how IL-1? induce CXCL10 expression in THP-1 monocytic cells, showing the evidence that IL-1? induces the expression of CXCL10 through activation of JNK/c-Jun signaling pathway. The pathophysiological context is that increased levels of IL-1?-induced CXCL10 could be associated with the development or severity of obesity and type-2 diabetes. Data is interesting but not novel. However, if CXCL10 is expected to play a key role in leukocyte homing to the inflamed tissue, my first concern is the lack of explanation for the use of a monocytic cell line. One of the most important immune, inflammatory cell involved in the pathophysiology of inflammatory diseases is the macrophage, and in particular Kupffer cells in the adipose tissue and the liver. The authors should justify why they did not induce differentiation of THP-1 cells into macrophages (via PMA or otherwise). This would have made it possible by better mimicking in vivo conditions and link the results with the pathophysiological context. They should also clearly describe the limitations of the findings with a cancerous model that resembles peripheral monocytes.
First, we appreciate the authors comments about the data being interesting. Regarding the specific question asked, we used THP-1 monocytic cell line because of homogeneity purposes because the global expression of genes, especially those involved in cellular activation, inflammation, immunity, and metabolism, might change during the transformation/ derivation of macrophages from monocytes or monocytic cells (doi: 10.1016/j.imlet.2007.12.012). We agree with the reviewer that macrophages are of pivotal importance in adipose and liver inflammation but in this preliminary study, we have focused on IL-1β-driven expression of CXCL10 in THP-1 monocytic cells. As suggested by the reviewer, the representative data from three independent experiments with similar results, showing IL-1?-induced CXCL10 expression in THP-1-derived macrophages are shown below. (These data are also included in the revised manuscript as supplemental material (Fig S1). As expected, THP-1-derived macrophages also show significant induction of CXCL10 secretory protein after stimulation with IL-1?, like THP-1 monocytic cells.
As the reviewer further suggested, we have also added in the limitations the point concerning use of a cancerous monocytic cell line/model in the study (Lines 300-302; in discussion section last paragraph).
Minor comments
Materials and Methods
1.The sequence of the primers used for qRT-PCR should be included in the manuscript, at least in the supplementary material.
Thanks for the suggestion. Accordingly, we have added the requested information in material methods. Since we purchased the primers from Thermo Scientific, we received TaqMan Gene Expression Assay product IDs. TaqMan Gene Expression Assay products (CXCL10 assay ID, Hs00171042_m1; c-Jun assay ID, Hs01103582_s1; JNK assay ID, Hs01548508_m1; and GAPDH assay ID, Hs03929097_g1) were purchased from Thermo Scientific.
- The catalogue numbers of the antibodies used for Western Blotting must be included.
Thank you. This information is now added to Western blotting section in Materials and Methods.
Results
Figure 2
- Authors discuss the decrease in CXCL10 mRNA levels with JNK inhibitor, but there seems to be an increase of mRNA levels with U0126 (MEK1/2 inhibitor) in particular, and probably also using PD98059 (ERK1/2 inhibitor). Moreover, it is mentioned that the other inhibitors “had a non-significant effect” (lines 96-97). This statement is regarding the inhibition of CXCL10 only, or is the increase also non-significant? I think it could be interesting to describe/analyse and discuss these results.
We appreciate the reviewer’s comments. As of increase in CXCL10 mRNA levels following treatments with U0126 or with PD98059, the changes were statistically non-significant as compared to treatments with other inhibitors used (SP600125 and SB203580). Regarding other comment about the statement spanning lines 96-97, it actually meant that only the inhibitor SP 600125 was able to suppress CXCL10 expression significantly at the transcriptional and translational levels while the other three inhibitors used (U0126, PD98059, and SB203580) induced a non-significant expression in CXCL10 at mRNA and protein levels, compared to IL-1β-induced CXCL10 expression without an inhibitor treatment. This sentence is now re-phrased for clarity.
- In the figure legend, the concentration of PD98059 is missing (line 100).
We apologize for this mistake. The ERK1/2 inhibitor PD98059 was also used at the concentration of 10 µM. The missing information is now incorporated as suggested.
Discussion
- Comment for figure 2 could be discussed.
Thanks for the suggestion. This comment is now added to the revised manuscript.

Reviewer 2 Report
Comments and Suggestions for Authors
Thank you for the opportunity to review your manuscript. The findings appear to be novel and may be of interest to the scientific community.
I have made a few suggestions regarding the structure of the manuscript. Please read through the attached pdf file and address the comments.

English is mostly excellent. There are, however, some instances of English language which would not usually be used. Although it is of a high standard, I would suggest having an English native speaker proof-read the document to make minor amendments.
Author Response
Response to Reviewer 2 Comments
Ref: Manuscript ID: pharmaceuticals-3004066
Title: IL-1β-induced CXCL10 expression in THP-1 monocytic cells involves the JNK/c-Jun and NF-κB mediated signaling
We thank the reviewer for thoughtful comments. Please see below the point-by-point responses to the comments or concerns raised.
Comments and Suggestions for Authors
Thank you for the opportunity to review your manuscript. The findings appear to be novel and may be of interest to the scientific community.
I have made a few suggestions regarding the structure of the manuscript. Please read through the attached pdf file and address the comments.
Author response: Thanks for your helping comments and corrections. All modifications are done as kindly suggested which has greatly improved the quality of the manuscript. All modifications from the original version have been shown using “track changes” tool in MS word.
Comments on the Quality of English Language
English is mostly excellent. There are, however, some instances of English language which would not usually be used. Although it is of a high standard, I would suggest having an English native speaker proof-read the document to make minor amendments.
Author response: Thanks for your highly valuable suggestion. The manuscript has been carefully re-read and edited by a native English speaker, with some changes as required.

Reviewer 3 Report
Comments and Suggestions for Authors
This study does not always present the novel findings in chemokine expression in diabesity since the mechanisms for CXCL10 expression have already been examined precisely. Further, there so many issues to be addressed as follows:
1.Why the authors noticed Cxcl10 expression in monocytic cells, not in other cell types should be explained in Introduction. For example, the CXCL10 expression in monocytic cells is increased in the pancreas of diabesity patients?
2.Is JNK involved in the activation of NFkB by IL-1b in monocytic cells? If JNK inhibitor suppresses phosphorylation of NFkB by IL-1b should be examined.
3.In Figure 2, p38 MAPK inhibitor or MEK inhibitor appeared to increase IL-1b-induced CXCL10 expression. The results indicate that p38MAPK and/or MEK/ERK might suppress the IL-1b-induced CXCL10 production. This point should be discussed.
4.I think that CXCL10 expression is dependent on STAT1. The authors should examine if STAT1 phosphorylation is enhanced by IL-1b as well as CXCL10 expression and if JNK inhibitor or NFkB inhibitor might suppress the IL-1b-induced STAT1 phosphorylation.
5.How the authors manipulate the results of this study should be presented. Can JNK or NFkB inhibitors be used as therapeutic agents for diabesity by suppressing CXCL10 expression in monocytic cells?
Comments on the Quality of English LanguageMinor English revision may be needed.
Author Response
Response to Reviewer 3 Comments
Ref: Manuscript ID: pharmaceuticals-3004066
Title: IL-1β-induced CXCL10 expression in THP-1 monocytic cells involves the JNK/c-Jun and NF-κB mediated signaling
We thank the reviewer for thoughtful comments. Please see below the point-by-point responses to the comments or concerns raised.
Comments and Suggestions for Authors
This study does not always present the novel findings in chemokine expression in diabesity since the mechanisms for CXCL10 expression have already been examined precisely. Further, there so many issues to be addressed as follows:
1.Why the authors noticed Cxcl10 expression in monocytic cells, not in other cell types should be explained in Introduction. For example, the CXCL10 expression in monocytic cells is increased in the pancreas of diabesity patients?
We thank the reviewer for the comment. Monocytes play a key role in metabolic inflammation in individuals with diabesity, and these individuals also display the increased circulatory levels of IL-1β and the chemokine IP10/CXCL10. Extravasation of activated monocytes from the circulation into the tissues (adipose tissue, liver, and pancreas) occurs during the disease progression and leads to inflammatory responses within these tissues of metabolic significance. However, it remains unclear whether IL-1β may play a direct role in the induction and/or upregulation of CXCL10 in monocytic cells. Therefore, we addressed the induction of CXCL10 in monocytic cells following stimulation with IL-1β. This point is now clarified in the Introduction section.
2.Is JNK involved in the activation of NFkB by IL-1b in monocytic cells? If JNK inhibitor suppresses phosphorylation of NFkB by IL-1b should be examined.
Thanks for the thoughtful comment. We have now tested whether the JNK inhibitor suppressed the phosphorylation of NF-κB by IL-1β and found that it did not, as shown below:
3.In Figure 2, p38 MAPK inhibitor or MEK inhibitor appeared to increase IL-1b-induced CXCL10 expression. The results indicate that p38MAPK and/or MEK/ERK might suppress the IL-1b-induced CXCL10 production. This point should be discussed.
We thank the reviewer for the comment. The non-significant increases in CXCL10 expression by using U0126 (MEK1/2 inhibitor), PD98059 (ERK1/2 inhibitor), and SB203580 (p38 inhibitor) are now mentioned in the discussion section, and the relevant references are also added.
4.I think that CXCL10 expression is dependent on STAT1. The authors should examine if STAT1 phosphorylation is enhanced by IL-1b as well as CXCL10 expression and if JNK inhibitor or NFkB inhibitor might suppress the IL-1b-induced STAT1 phosphorylation.
Thanks for the comment. As suggested, we have tested whether IL-1β induced STAT1 phosphorylation and found that it did not (IFN-γ stimulation was used as a positive control). These data are shown below.
5.How the authors manipulate the results of this study should be presented. Can JNK or NFkB inhibitors be used as therapeutic agents for diabesity by suppressing CXCL10 expression in monocytic cells?
Thanks for the useful comments, which have now been addressed in the Discussion section.
Comments on the Quality of English Language
Minor English revision may be needed.
The manuscript text has now been revised as well as carefully edited by a native English speaker for necessary linguistic corrections.
Submission Date
29 April 2024
Date of this review
15 May 2024 06:52:36

Round 2
Reviewer 3 Report
Comments and Suggestions for Authors
The authors well addressed the issues I pointed out and appropriately revised the article.